# Role of Alcohol Drinking in Alzheimer’s Disease, Parkinson’s Disease, and Amyotrophic Lateral Sclerosis

**DOI:** 10.3390/ijms21072316

**Published:** 2020-03-27

**Authors:** Bin Peng, Qiang Yang, Rachna B Joshi, Yuancai Liu, Mohammed Akbar, Byoung-Joon Song, Shuanhu Zhou, Xin Wang

**Affiliations:** 1Departments of Neurosurgery, Brigham and Women’s Hospital, Harvard Medical School, Boston, MA 02115, USA; 2Hubei Provincial Key Lab for Quality and Safety of Traditional Chinese Medicine Health Food, Jing Brand Research Institute, Daye 435100, China; 3Internal Medicine, Stafford Medical, PA. 1364 NJ-72, Manahawkin, NJ 08050, USA; 4Division of Neuroscience & Behavior, Laboratory of Membrane Biochemistry and Biophysics, National Institute on Alcohol Abuse and Alcoholism, National Institutes of Health, Rockville, MD 20852, USA; mohammed.akbar@nih.gov; 5Section of Molecular Pharmacology and Toxicology, Laboratory of Membrane Biochemistry and Biophysics, National Institute on Alcohol Abuse and Alcoholism, National Institutes of Health, Rockville, MD 20892, USA; bj.song@nih.gov; 6Departments of Orthopedic Surgery, Brigham and Women’s Hospital, Harvard Medical School, Boston, MA 02115, USA

**Keywords:** alcohol, neurodegenerative diseases, Alzheimer’s disease, Parkinson’s disease, Amyotrophic lateral sclerosis

## Abstract

Neurodegenerative diseases, including Alzheimer’s disease (AD), Parkinson’s disease (PD) and amyotrophic lateral sclerosis (ALS), increase as the population ages around the world. Environmental factors also play an important role in most cases. Alcohol consumption exists extensively and it acts as one of the environmental factors that promotes these neurodegenerative diseases. The brain is a major target for the actions of alcohol, and heavy alcohol consumption has long been associated with brain damage. Chronic alcohol intake leads to elevated glutamate-induced excitotoxicity, oxidative stress and permanent neuronal damage associated with malnutrition. The relationship and contributing mechanisms of alcohol with these three diseases are different. Epidemiological studies have reported a reduction in the prevalence of Alzheimer’s disease in individuals who drink low amounts of alcohol; low or moderate concentrations of ethanol protect against β-amyloid (Aβ) toxicity in hippocampal neurons; and excessive amounts of ethanol increase accumulation of Aβ and Tau phosphorylation. Alcohol has been suggested to be either protective of, or not associated with, PD. However, experimental animal studies indicate that chronic heavy alcohol consumption may have dopamine neurotoxic effects through the induction of Cytochrome P450 2E1 (*CYP2E1*) and an increase in the amount of α-Synuclein (αSYN) relevant to PD. The findings on the association between alcohol consumption and ALS are inconsistent; a recent population-based study suggests that alcohol drinking seems to not influence the risk of developing ALS. Additional research is needed to clarify the potential etiological involvement of alcohol intake in causing or resulting in major neurodegenerative diseases, which will eventually lead to potential therapeutics against these alcoholic neurodegenerative diseases.

## 1. Introduction and Alcohol Use: Dual Effects and Mechanisms

As the population ages, we need to pay more attention to age-related neurodegenerative diseases, including Alzheimer’s disease (AD), Parkinson’s disease (PD) and Amyotrophic lateral sclerosis (ALS) [1]. These neuronal diseases are chronic and progressive, decreasing the quality of life for both patients and their caregivers and bringing a heavy economic burden to society [2]. A small number of cases are caused by known genetic mutations, and studying these variants would help our understanding of the pathogenesis of these neurodegenerative disorders [3]. However, environmental factors, acting as susceptibility factors and/or triggers, play an important role in most cases [4].

Several environmental factors are associated with these neurodegenerative diseases [5,6,7]. The main risk factors for AD, PD and ALS are listed in Table 1. Alcohol consumption exists extensively throughout the world. Complicated genetic and environmental factors are likely to determine the pathophysiological consequences of alcohol drinking [8]. The relationship between alcohol intake with various disorders has been studied for decades [9]. Mild to moderate regular consumption might be beneficial [10]. However, chronic and excessive alcohol drinking leads to social, economic and multi-organ problems, including an altered neurological system [11], liver disease [12], cardiotoxicity [13], psychiatric disorders [14], cancer [15], and endocrine [16], microbiome [14] and immunity disruption [17].

Chronic alcohol consumption has a pronounced impact on humans through complicated mechanisms (Figure 1). Ethanol plays a harmful role by affecting neurotransmitter systems in a variety of ways. Ethanol stimulates inhibitory γ-aminobutyric acid (GABA) receptors and suppresses excitatory glutamate receptors [20]. After drinking, alcohol can cause brief euphoria, then progresses to respiratory depression and coma in a dose-dependent manner [21]. Moreover, chronic alcohol intake leads to increased glutamate-induced excitotoxicity, oxidative stress [22] and permanent neuronal damage associated with malnutrition, such as dementia with decreased cognitive and executive function, cerebellar degeneration with impaired motor skills and/or ataxia, polyneuropathy and Wernicke and Korsakoff syndromes [23]. Ethanol also negatively affects the availability of nerve growth factor and brain-derived neurotrophic factor, resulting in impaired intracellular signaling pathways [23]. However, on the other hand, some studies show that moderate amounts of ethanol intake produce a neuroprotective effect. In addition, alcohol can increase insulin sensitivity [24], stimulate fibrinolysis [25], oppose thrombin activity [26], prevent platelet aggregation [27] and reduce inflammatory markers [28].

There is accumulating evidence that alcohol intake changes the microbiota and the microbiota–gut–brain axis [14]. Alcohol-induced alterations of the microbiome can cause neuroinflammation and alter the balance of the neuroimmune function. On the other hand, excessive amounts of alcohol interact with neurotransmitter systems and increase blood–brain barrier (BBB) permeability, resulting in brain damage and dysfunction [20]. The gut microbiota plays an important role in the pathogenesis of neurodegenerative disorders. Recent research shows that probiotics may improve the symptoms of neurodegenerative disorders, suggesting a causal link between gastrointestinal alterations and the etiology of many neuronal diseases, including AD, PD and ALS [29].

Depending on the amounts of intake, alcohol may have dual roles in worsening or in protecting neurodegenerative diseases (Figure 2). Epidemiological studies have reported a reduction in the prevalence of AD in individuals who ingest low amounts of alcohol [30]. Animal and cell studies show that low or moderate concentrations of ethanol reduce the damage induced by β-amyloid (Aβ) in AD [30,31,32,33]. Higher concentrations of ethanol contribute to increased accumulation/production of Aβ and the precursor protein [34]. One recent study showed that alcohol drinking in the triple-transgenic model of AD (3xTg-AD) mice induced deficits in cognitive and emotional functions compared to wild-type controls, and induced pathological changes such as the hyperphosphorylation of Tau-Ser199/202 in neuronal cell bodies and the dorsal hippocampus (CA1) in 3xTg-AD mice 1 month post alcohol drinking [35]. Alcohol has been suggested to be either protective of, or not associated with PD [36]. However, experimental animal studies indicate that chronic heavy alcohol consumption may have dopamine neurotoxic effects [37]. Chronic alcohol exposure decreased the levels of dopamine (DA) [38] and increased the amount of α-Synuclein (αSYN), which was linked to the development of PD [39]. The findings in the association between alcohol consumption and ALS are inconsistent; a recent population-based study suggests that drinking alcohol does not seem to influence the risk of developing ALS [40]. One animal study showed that red wine extract protects cells from glutamate-induced apoptosis and extends the lifespan of the transgenic mice expressing the human mutated Cu, Zn superoxide dismutase (mSOD1) [41]. However, the relationship between alcohol and ALS remains to be established.

Despite much progress being made in this area, there are still some uncertain questions that need to be discussed. In this article, we review the main neurological effects of alcohol intake on neurological disorders and describe the epidemiological and experimental relationships between alcohol consumption and major neurodegenerative diseases such as AD, PD and ALS.

## 2. Dual Roles of Alcohol Intake in Alzheimer’s Disease (AD) Development and Progression

### 2.1. Evidence from Epidemiological Studies

With continuously increasing life expectancy, cognitive impairment and the global burden of AD is increasing. Worldwide, approximately 50 million people have dementia in 2019 and, with one new case every three seconds, the number of people with dementia is expected to triple by 2050 [43]. AD is one of the most common diseases within the elderly population and is the leading cause of dementia, accounting for 60–70% of all cases [5]. 

Many factors contribute to dementia. Some factors are nonmodifiable, such as family history, aging, head injury, and genetic determinants, e.g., carrying the epsilon 4 allele of the apolipoprotein E (*APOE4*) gene. Some factors are modifiable, such as body mass index (BMI), elevated blood pressure with vascular dysfunction, stroke, diabetes mellitus, elevated total cholesterol levels, current smoking, poor diet, depression, cognitive inactivity, physical inactivity, and low educational attainment. These risk factors could serve as potential targets for the prevention of dementia [44]. 

Studies have shown a relationship between the development of cognitive impairment and dementia with the harmful use of alcohol [43]. However, the association of alcohol consumption with cognitive outcomes is controversial. Observational studies suggest that heavy alcohol consumption is related to clinical alcoholic dementia and leads to deterioration of cognitive and executive function, while moderate alcohol intake may have a protective effect. The relationship between alcohol consumption and cognitive decline is thought to be J-shaped or U-shaped [43].

A 2019 meta-analysis reviewed 91 articles involving 36 risk factors for AD [45]. The authors concluded that significant associations with AD decreased risk were found for alcohol consumption (any or light/moderate versus abstinence) from risk ratio (RR) = 0.43 (95% CI = 0.17–0.69) [46] to 0.72 (95% CI = 0.61–0.86) [47]. Anstey et al. [47] presented a meta-analysis of 15 prospective studies, concluding that drinkers had a lower risk of AD than nondrinkers (RR = 0.66, 95% CI = 0.47–0.94). However, there were not many studies focused on whether the benefits of alcohol intake last throughout adulthood or if a specific protective effect of alcohol shows up in the latter part of life. A prospective population-based study [48] found that alcohol intake in middle age revealed a U-shape relationship with cognitive declines and that participants who drank infrequently had the lowest likelihood of mild cognitive impairment in old age in comparison to the participants who did not drink or drank frequently. This study suggests that an increased risk of dementia with greater alcohol intake was only associated with people carrying the *APOE4* gene. 

It seems that the effect of alcohol on AD is related to the amount, pattern and frequency of drinking, the type of alcohol, the nutritional state of individuals and the human genotype. Sabia et al. [49] reported the Whitehall II cohort study that examined the link between alcohol consumption and the risk of dementia in a 23-years-long follow-up study, specifically, with a focus on the amount of alcohol intake per week. They concluded that the risk of dementia was increased in people who abstained from alcohol in midlife or consumed >14 units/week. Furthermore, results from one MRI study [50] showed that wine consumption was related to less white matter lesions and better cerebral blood flow, demonstrating better cognitive function. Stampfer et al. [51] published a cohort study examining the association between moderate alcohol intake and cognitive ability in women, concluding that less than one drink per day may be related to a lower risk of cognitive impairment.

Although evidence indicates that moderate alcohol consumption decreases the risk of AD, several studies claim that moderate alcohol consumption promotes harmful effects. One 2.5–4 years follow-up study [52] found no association between low-to-moderate alcohol consumption and cognitive impairment. Furthermore, Toda et al. [53] presented their study involving AD patients with a habit of drinking and suggested that the cognitive decline of AD patients was reduced after the patients stopped drinking alcohol. 

The evidence-based studies of alcohol consumption and AD are listed in Table 2. Some research results regarding alcohol and dementia or cognitive decline, which did not discuss AD, were excluded. Most studies support the J-shaped or U-shaped relationship between alcohol consumption and the risk of AD. Due to methodological limitations in most studies regarding the U-shaped relationship between alcohol consumption and cognitive function, in addition to other health risks and the social and economic burden associated with alcohol, it is not possible to assume that light-to-moderate consumption of alcohol is, in fact, protective toward dementia and/or cognitive decline; thus, the WHO guidelines in 2019 do not favor a general recommendation of its use [43].

### 2.2. Studies in Animal and Cell Culture Models

The pathogenesis of AD is related to two proteins in the brain, Aβ and microtubule-associated protein tau, that proliferate, aggregate and get deposited in the brain, leading to memory decline and causing behavioral changes. Bate et al. [62] reported that low concentrations of ethanol (0.02–0.08%) partially protect cultured cortical and hippocampal neurons against the Aβ-induced synapse toxicity. The results are similar to another observation that low concentrations of ethanol protect against damage induced by Aβ in primary neuronal cultures [30,31] and cell lines [32]. Moreover, they found that after consuming 6% ethanol for 2 months, the learning and memory abilities in 3xTg-AD mice were improved [30]. In another study [33] employing moderate ethanol preconditioning (20–30 mM for 6 days) of organotypic hippocampal-entorhinal slice cultures, it was shown that ethanol had a protective effect on Aβ-induced damage (i.e., neurotoxicity and apoptosis).

Huang et al. [34] used neuronal cell cultures, which were treated with ethanol at different concentrations, and AD-model mice, which were exposed to 20% (*v*/*v*) ethanol for 4 weeks, and showed that the Aβ precursor protein increased both in vitro and in vivo, indicating that excessive ethanol promoted the pathogenesis of AD. Interestingly, glycogen synthase kinase-3 (GSK3) is not only critical to the hyperphosphorylation of Tau and related to Aβ induced cell death, but it also regulates the effects of alcohol. GSK3β phosphorylation is increased after acute and chronic alcohol exposure. One recent study showed hyperphosphorylation at the GSK3 site on Tau protein in the hippocampus (CA1) subregion of a 3xTg-AD mouse brain 1 month post alcohol drinking, suggesting that chronic alcohol drinking has detrimental effects in AD [35].

Overall, evidence from AD animal models and cell cultures support the idea that low or moderate concentrations of ethanol exhibit a protective effect in vitro and in vivo. In contrast, higher concentrations of ethanol intake promote AD pathogenesis. The animal studies are consistent with the epidemiological data.

## 3. Dual Roles of Alcohol Intake in Parkinson’s Disease (PD) Development and Progression

### 3.1. Evidence from Epidemiological Studies

Another age-related common neurodegenerative disease, Parkinson’s disease (PD), is characterized by a constellation of clinical manifestations, which include slowness of movement, rest tremor, rigidity and postural instability [63]. Its cause remains unclear in most cases. The prevalence and incidence of PD, which is predominantly observed in males, increase with age [64]. 

Several epidemiological studies have assessed the relationship between environmental factors and the risk of developing PD [18,65]. Some factors are likely to increase the risk, such as exposure to pesticides [66], traumatic brain injury [67], history of melanoma [68], consumption of dairy products [69] and high iron intake [70]. Reduced risks are associated with physical activity [71], mild smoking [72], tea [73], vitamin E [74], caffeine consumption [73], higher serum urate concentrations [75] and the use of ibuprofen and other common medications [76]. Smoking and caffeine consumption, two common daily behaviors, have been consistently associated with a reduced risk of PD. However, the dual roles of positive and negative results from epidemiological studies on alcohol intake and PD risk have been reported [36,37,77,78].

Lifestyle factors, including caffeine or alcohol consumption, physical activity or cigarette smoking may contribute to the development of Parkinson’s disease. Relative to moderate drinkers, those who never drank liquor and those who drank more heavily were at an increased risk of Hoehn and Yahr 3 (hazard ratio, 3.48; 95% confidence interval, 1.90–6.38; and hazard ratio, 2.16; 95% confidence interval, 1.03, 4.54, respectively) [79]. Zhang et al. [36] published a meta-analysis of the observational studies regarding the link between alcohol intake and PD. From 32 studies, including 677,550 subjects, they concluded that beer (RR = 0.59, 95% CI: 0.39–0.90), but not wine and liquor, probably protected against PD, especially for males (RR = 0.65, 95% CI: 0.47–0.90) but not for females. However, there were not enough studies focused on dose-response analysis and the relationships between beer, wine and liquor. Paganini-Hill [77] published the Leisure World Cohort Study: risk factors for PD. This case-control study investigated several risk factors such as smoking, alcohol, coffee, hypertension and vitamins. They found that drinkers who consumed 2 or more drinks of alcoholic beverages/day had a lower risk of PD than abstainers (OR =0.73, 95% CI, 0.56–0.96), as well as consuming wine (OR = 0.80, 95% CI: 0.51–1.24), beer (OR = 0.32, 95% CI: 0.10–1.06) and hard liquor (OR = 0.75, 95% CI: 0.55–1.01). Therefore, they concluded that the risk of PD was reduced among alcohol consumers. 

Palacios et al. [78] reported the Cancer Prevention Study (CPS) II, Nutrition Cohort Study from 1992 to 2005, a large prospective cohort to assess whether alcohol consumption was associated with the risk of PD. After adjustment for several risk factors, the relative risk of men consuming alcohol (30 or more grams/day) compared to non-drinking men was 1.29 (95% CI: 0.90–1.86) while the relative risk of women consuming alcohol (15 or more grams/day) relative to non-drinking women was 0.77 (95% CI: 0.41–1.45). In addition, consumption of beer, wine or liquor was not related to PD risk in the CPS-II Nutrition cohort. Their general conclusion, which showed no support for the relationship between total alcohol intake and the incidence of PD, contradicts other studies. In a study based on the Swedish National Inpatients Register and including over 1000 cases of PD [37], chronic heavy alcohol intake increased the risk of PD.

The main evidence-based reviews on the relationship between alcohol intake and PD are listed in Table 3. Among the reports on the relationship between alcohol and these three neurodegenerative diseases, more research has been done on PD. Unfortunately, perhaps due to the uncertainty, these study results are not consistent. The relationship between alcohol drinking and PD is complex and needs further study.

### 3.2. Studies with Animal and Cell Culture Models

PD is characterized by the loss of dopamine neurons in the substantia nigra pars compacta (SNpc) and later in the ventral tegmental area, with progressive loss of DA activity in the nucleus caudate and putamen, involved primarily in motor regulation and movement disabilities [93,94,95,96,97,98]. Ethanol has been shown to produce neurotoxicity in different brain regions of mice, including cerebral cortices, the hippocampus and the cerebellum [99]. Some reported alcoholics develop parkinsonism, a group of neurological disorders that cause movement problems similar to those seen in Parkinson’s disease, after alcohol withdrawal [100]. Chronic alcohol intake and dopaminergic signaling have a potential relationship [101]. Chronic alcohol exposure decreases levels of DA and 3, 4-dihydroxyphenylacetic acid (DOPAC) in the ventral striatum, as well as tyrosine hydroxylase (TH) protein levels. Another report found that DA and its metabolites were depleted after alcohol exposure [38]. In addition, alcohol decreases dopaminergic neurons, possibly through induction of Cytochrome P450 2E1 (*CYP2E1*), one of the major enzymes in the central nervous system (CNS) that catalyzes ethanol oxidation [102]. The CYP2E1 response induced by ethanol is one of its important metabolic pathways. The CYP2E1 enzyme can metabolize ethanol to acetaldehyde, a highly reactive and toxic compound, which can enhance the 1-methyl-4-phenyl-1,2,3,6-tetrahydropyridine (MPTP) -induced parkinsonism in mice [103]. The lesions were significantly reduced in the substantia nigra of *CYP2E1* knockout mice treated with neurotoxin MPTP compared to lesions observed in wild-type animals [103]. Moreover, in the brain of Parkinson’s patients, decreased methylation of the *CYP2E1* gene with increased *CYP2E1* mRNA levels were observed, suggesting that epigenetic variations of *CYP2E1* contribute to PD susceptibility [104]. 

αSYN is the neuropathological protein that is related to and is likely to contribute to the development of PD. Accumulation of αSYN could be increased under oxidative stress [105], which can be produced by increased *CYP2E1* following exposure to alcohol [106] or MPTP [107], which is frequently used to develop animal models of PD [108]. Moreover, the gene encoding αSYN (*SNCA*) is identified as one of the most important genetic contributors to both familial and sporadic PD. Interestingly, genetic studies show that *SNCA* is also linked to alcoholism. Alcohol-preferring rats, compared to non-alcohol preferring counterparts, exhibit reduced DA overflow and contain a higher level of αSYN in the nucleus accumbens core. Therefore, αSYN might affect dopaminergic neurotransmission and is linked to alcohol-related altered behaviors and movement disorders [39]. 

However, it is well established that many antioxidants, including various flavonoids contained in fruits and vegetables, can show neuroprotective effects against cell and animal models of PD [109]. Yu et al. [110] reported that ethanol at low doses had protective effects against methamphetamine-induced dopamine depletion. They concluded that the multiple effects of ethanol could probably be associated with nonspecific receptor systems, resulting in neurotoxicity. Another study showed that the presence of other beneficial agents such as resveratrol contained in red wine might mitigate the toxicity of methamphetamine [111]. Therefore, evidence from PD animal models and cell cultures is inconsistent. Similar conclusions were drawn in epidemiological studies. To date, a categorical association between alcohol and PD has not been clearly established. Therefore, further research is needed before a clear conclusion can be drawn.

## 4. Dual Roles of Alcohol in Amyotrophic Lateral Sclerosis (ALS) Development and Progression

### 4.1. Evidence from Epidemiological Studies

Amyotrophic lateral sclerosis (ALS) is a highly progressive and fatal motoneuron disease characterized by weakness of muscles, leading to increased muscle wasting, dysarthria, dysphagia and respiratory failure [112,113,114]. The prevalence and incidence in males is higher than females [115]. Extending life expectancy in ALS seems to be dependent on improving our understanding of its pathogenesis mechanisms. About 10% of ALS patients are familial, with a Mendelian pattern of inheritance. Sporadic ALS is one of the most devastating neurological diseases; most patients die within 3–4 years after symptom onset. ALS is affected by both genetic and environmental factors. Mutations in genes such as *TDP-43*, *SOD1*, *C90RF72 A* and *FUS* may explain 60–80% of familial ALS [116]. 

Autopsy analyses and laboratory studies on ALS indicate that oxidative stress plays a major role in motor neuron degeneration and astrocyte dysfunction [117]. Oxidative stress biomarkers are elevated in cerebrospinal fluid, plasma and urine, suggesting that abnormal oxidative stress is generated outside of the CNS. The CNS is particularly susceptible to oxidative stress because the neuronal membranes contain a high abundance of polyunsaturated fatty acids, especially arachidonic and docosahexaenoic acids. Post mortem analysis of neuronal tissues of patients with sporadic ALS consistently shows oxidative damage to lipids, proteins and DNA. 

Given that only 10% of all ALS cases are familial and 90% are sporadic, environmental factors may play a greater role in the disease progression. Epidemiological studies show geographical prevalence and incidence [118]. It seems that developed regions have a higher ALS prevalence and incidence compared to less-developed areas. Based on the gender differences observed in ALS patients, several groups have reported that estrogen and hormonal mechanisms may also play a potential role [119]. The effects of various environmental factors, including smoking [120], trauma [121], physical activity [122], pesticides [123] and heavy metals [124] have been investigated; smoking and pesticides are potential risk factors. 

The findings on the association between alcohol consumption and ALS are inconsistent; a recent population-based study suggests that drinking alcohol seems to have no influence on the risk of developing ALS [40]. Due to a mean incidence of 1–2 cases of ALS per 100,000 people per year, the cohort and case-control studies are very limited and thus it is difficult to establish a consensus [125]. Despite the limited number of case studies, Meng et al. [126] analyzed data from different types of epidemiological studies, including one cohort study and seven case-control studies. The resulting meta-analysis [126] suggested that the risk of developing ALS was reduced among alcohol consumers (OR = 0.57, 95% CI 0.51–0.64), indicating that alcohol drinking has a protective effect on the development of ALS. However, there was only one study that examined the dose-response relationship between alcohol intake and ALS, so this relationship was not very clear. 

Sonja et al. [127] presented a population-based study regarding the association between smoking, alcohol intake and ALS. Both former drinkers (OR = 0.67, 95% CI: 0.40–1.13) and current drinkers (OR = 0.52, 95% CI: 0.40–0.75) are associated with a lower risk of ALS, indicating that alcohol intake has an inverse association with the risk of ALS. With few exceptions, smoking increased the risk of ALS (OR = 1.38, 95% CI: 1.02–1.88). Another case-control study [128] published in 2015 also drew a similar conclusion that higher intake of alcohol protected against ALS (OR = 0.91, 95% CI: 0.84–0.99). However, the 2019 Euro-Motor study, focusing on the relationship between alcohol consumption and the risk of ALS, reported that alcohol drinking (OR = 0.93, 95% CI: 0.75–1.15) and red wine drinking had no significant association with the risk of ALS. At least, alcohol intake was not indicated as a higher risk of ALS [40].

The main evidence-based reviews on the relationship between alcohol intake and ALS are listed in Table 4. Weak evidence suggests that alcohol consumption may reduce the risk of ALS. These studies imply that alcohol drinking has a protective effect or that there was no strong association between alcohol intake and the risk of ALS, indirectly indicating alcohol drinking did not facilitate the development of ALS. It is, therefore, possible that alcohol intake may weakly protect against ALS.

### 4.2. Studies with Animal and Cell Culture Models

The animal models of ALS can be prepared by a variety of genetic mutations [134]. These ALS mice or rats show signs of age-dependent progressive muscle weakness. These rodents are more susceptible to severe neuromuscular damage and deaths than the age- and sex-matched wild-type counterparts. ALS rodents show signs of increased oxidative stress. Since alcohol intake is also known to increase oxidative and nitrative stress, alcohol consumption is likely to accelerate the development and progression of ALS animals in an additive or synergistic manner. 

Antioxidants may play a protective role, since lyophilized red wine extract prolonged the lifespan of mSOD1 transgenic mice [41]. In recent years, laboratory studies have revealed that astrocytic glial cells play a key role in neuronal degeneration. When cultured astrocytes undergo oxidative damage, glutamate transport is impaired, probably resulting in excitotoxic neuronal injury [135]. Emerging evidence indicates that there is a dysfunction of the glutamatergic excitatory system in ALS, and chronic alcohol intake involves changes in glutamatergic transmission [136]. In addition, several drugs such as ceftriaxone, an FDA-approved drug, which can elevate the glutamate transporter expression, not only reduce the alcohol intake as a treatment of alcohol-preferring rats but are also currently in clinical trials for the treatment of ALS [136]. In this consideration, it is likely that alcohol consumption probably accelerates the development and progression of ALS animals. However, we cannot come to a conclusion, as there are too few animal experiments on alcohol intake and ALS.

## 5. Conclusions

In summary, we have reviewed recent epidemiological and experimental evidence regarding the association between alcohol consumption and the development of age-related neurodegenerative diseases. There are some inconsistent results in these studies. Age is not only a leading risk factor for the development of these neurodegenerative diseases, but also implies potential additive or synergistic effects on exposure to habitual behaviors such as chronic alcohol consumption. In addition, exposure to environmental risk factors such as binge alcohol drinking and habitual unhealthy behaviors like smoking or substance abuse could play important roles in the development and progression of neurodegenerative diseases. Alcohol consumption is one of the most common habitual behaviors worldwide; however, it also represents a preventable habit by a behavioral change of simple abstinence. Several mechanisms exist that might explain how the history and/or pattern of alcohol intake could increase or decrease the risk of developing various neurodegenerative diseases. The relationship and contributing mechanisms of alcohol toward these three diseases are different. Epidemiological studies report a reduction in the prevalence of AD in individuals who drink low amounts of alcohol; low or moderate concentrations of ethanol protect against Aβ toxicity in hippocampal neurons; and excessive amounts of ethanol increase accumulation of Aβ and Tau phosphorylation, leading to neuronal cell death and neurodegeneration. Alcohol drinking has been suggested to be either protective of, or not associated with, the development of PD. However, experimental animal studies indicate that chronic heavy alcohol consumption may have dopamine neurotoxic effects through the induction of *CYP2E1* and increase in the amount of αSYN relevant to PD. The findings in the association between alcohol consumption and ALS are inconsistent; a recent population-based study suggests that alcohol consumption seems to have no influence on the risk of developing ALS. Further research is needed to clarify the potential etiological involvement of alcohol intake in causing or resulting in major neurodegenerative diseases, which will eventually lead to potential therapeutics against these neurodegenerative diseases.

## Figures and Tables

**Figure 1 ijms-21-02316-f001:**
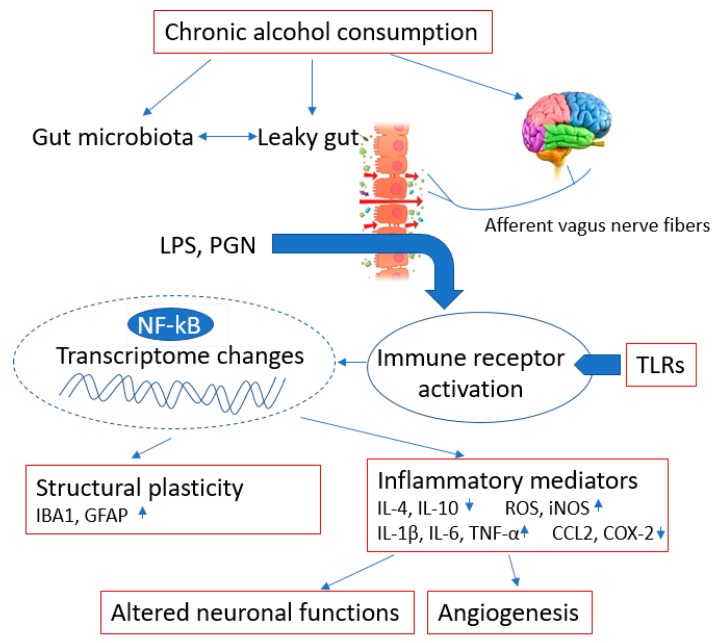
The harmful effects and mechanisms of alcohol. Chronic and excessive alcohol consumption alters gut microbiota, increases intestinal permeability (leaky gut) and causes brain damage. Lipopolysaccharide (LPS) and peptidoglycan (PGN) from the gut lumen to the systemic circulation are recognized by toll-like receptors (TLRs) expressed by immune cells and induce structural plasticity changes such as ionized calcium-binding adapter molecule 1 (IBA1) and glial fibrillary acidic protein (GFAP), as well as an inflammatory response by the nuclear factor-ĸB (NF-ĸB) pathway. The inflammatory cytokines and mediators, such as interleukin 4 (IL-4), IL-10, reactive oxygen species (ROS), inducible NO synthase (iNOS), IL-1β, IL-6, tumor necrosis factor (TNF)-α, C-C Motif Chemokine Ligand 2 (CCL2) and cyclooxygenase-2 (COX-2) may result in altered neuronal functions and angiogenesis.

**Figure 2 ijms-21-02316-f002:**
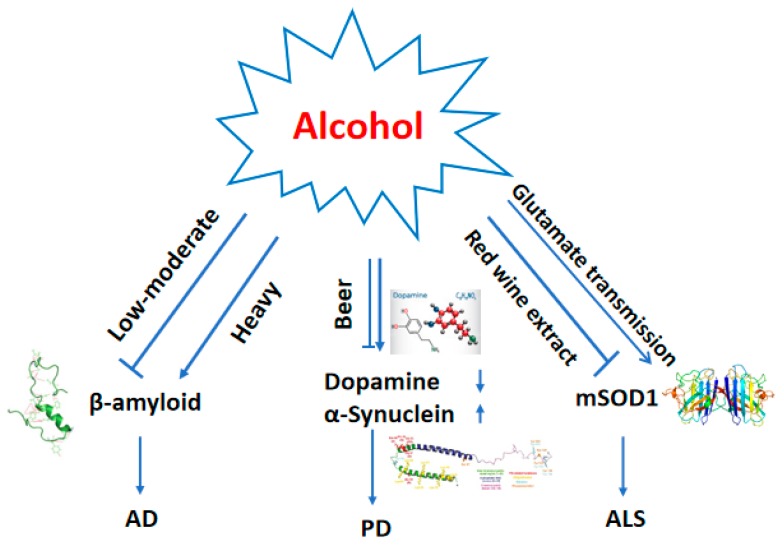
Dual roles of alcohol drinking in Alzheimer’s disease (AD), Parkinson’s disease (PD) and Amyotrophic lateral sclerosis (ALS). In AD, low or moderate concentrations of ethanol protect against the damage induced by Aβ. Excessive amounts of ethanol increase accumulation of Aβ and Tau phosphorylation [35]. In PD, most studies show that alcohol decreases levels of DA and increases the levels of αSYN. One study suggested that the high drinking alko alcohol (AA) rats showed lower stimulated DA levels and more efficient DA re-uptake in the nucleus accumbens core than the low drinking non-alcohol (ANA) rats did. In the same structure, AA rats also had significantly higher levels of α-syn. No differences were found in the effects of two different doses of ethanol (0.1 and 3.0 g/kg) [42]. A meta-analysis concluded that beer, but not wine and liquor, probably protected against PD [36]. In ALS, red wine extract intake was reported to prolong the lifespan of mSOD1-transgenic mice and protected neurons from apoptosis in vitro. There is a dysfunction of the glutamatergic excitatory system in ALS, and chronic alcohol intake involves changes in glutamatergic transmission. The association between alcohol and ALS is still unclear. Abbreviations: Aβ: β-amyloid; DA: Dopamine; mSOD1: Mutated SOD1.

**Table 1 ijms-21-02316-t001:** Important risk factors for Alzheimer’s disease (AD), Parkinson’s disease (PD) and Amyotrophic lateral sclerosis (ALS).

	AD	PD	ALS
Increased risk, ↑ robust evidence	Pesticides, elevated blood pressure, elevated total cholesterol levels, current smoking, head injury, *APOE4* gene family history, aging	Pesticides, aging	Smoking, pesticides, family history, aging, gender
Increased risk, ↑ weak evidence	BMI, diabetes mellitus, alcohol (excessive), poor diet, depression	Consumption of dairy products, history of melanoma, traumatic brain injury, high iron intake, chronic anemia, alcohol	Head trauma, low premorbid BMI, workers or farmers, heavy metals, BMAA, previous viral infection, electrical magnetic fields, strenuous physical activity
Reduced risk, ↓ robust evidence	Alcohol (moderate), physical exercise, cognitive activity, educational attainment	Smoking, caffeine consumption, higher serum urate concentrations [18]	
Reduced risk, ↓ weak evidence	Mediterranean diet, coffee, non-steroidal anti-inflammatory drugs	Physical activity, ibuprofen and other common medications, tea, Vitamin E	Vitamin E, high blood pressure, longer duration of education, alcohol, reading, retirement, hyperglycemia [19]

*APOE4**:* epsilon 4 allele of the apolipoprotein E; BMI: Body mass index; BMAA: β-methylamino-L-alanine.

**Table 2 ijms-21-02316-t002:** Evidence-based reviews on the relationship between alcohol intake and AD.

Reference, Year	Study Population	Study Design	No. of Subjects(Cases/Controls)	Diagnosis Criteria	Adjusted Confounders	Alcohol Exposure	OR (95%CI)
Langballe et al., 2015 [54]	Population-based	Cohort	595/40,435	ICD-10	Age, sex, years of education, hypertension, obesity, smoking, and symptoms of depression	Abstainers; Drinking 0 times, not abstainers; Drinking 5 or more times; Unknown	1.09 (0.80–1.48) 1.20 (0.96–1.51) 1.47 (1.00–2.16) 1.73 (1.17–2.55)
Weyerer et al. 2011 [55]	Prospective longitudinal study	Cohort	111/3202	DSM-III-R, DSM-IV and ICD-10	Age, gender, education, living situation, IADL impairment, somatic co-morbidity, depression, *APOE4*, MCI and smoking	Abstinent; Not abstinent; Wine (only); Beer (only); Mixed (wine, beer and other alcoholic beverages)	1 (referent group) 0.58 (0.38–0.89) 0.76 (0.46–1.23) 0.60 (0.30–1.21) 0.14 (0.03–0.56)
Zhou et al., 2011 [56]	Population-based	Cohort	172/3170	DSM-IV Criteria	Age, BMI, education, *APOE4*, vascular risk Factors	Occasional drinking; Monthly drinking; Weekly drinking; Daily drinking	1 1.03 (0.83–1.35) 1.31 (0.69–1.43) 2.25 (1.43–3.97)
Luchsinger et al., 2004 [57]	Cohort of elderly persons	Cohort	199/2126	DSM-IV Criteria	Age, gender, education, *APOE4*, Heart disease	None; Light to moderate Beer Liquor Wine	1 1.33 (0.91–1.96) 1.15 (0.77–1.71) 0.59 (0.38–0.91)
Mukamal et al. 2003 [58]	Cohort Community-based	Case- control	373/373	DSM-IV Criteria	Age, sex, race, *APOE4*, education, income, marital status, estrogen replacement treatment, smoking, DM-2, BMI, total cholesterol level, atrial fibrillation, heart disease, stroke, TIA, physical activity	Less than 1 drink/week; 1–6 drinks/week; 7–13 drinks/week; 14 or more drinks/week	0.65 (0.41–1.02) 0.46 (0.27–0.77) 0.69 (0.37–1.31) 1.22 (0.60–2.49)
Lindsay et al. 2002 [59]	Nationwide, population-based	Cohort	194/3894	DSM-IV Criteria	Age, sex, education, family history of dementia, *APOE4,* arthritis, NSAIDs, wine consumption, coffee consumption, regular physical activity	At least weekly consumption of Beer; Wine; Spirits; Alcohol (any type)	0.84 (0.51–1.41) 0.49 (0.28–0.88) 0.78 (0.52–1.19) 0.68 (0.47–1.00)
Ruitenberg et al., 2002 [60]	Population-based	Cohort	146/7983	DSM-III-R Criteria	Age, sex, BMI, SBP, diabetes, smoking, and education.	<1 drink per week; ≥1 drink per week but <1 per day; 1–3 drinks per day; ≥4 drinks per day	0.91 (0.58–1.44) 0.91 (0.58–1.44) 0.72 (0.43–1.20) 1.17 (0.35–3.55)
Huang et al., 2002 [61]	Community-based dementia- free cohort	Cohort	84/402	DSM-III-R criteria	Age, gender, education, smoking, institutionali-zation	Nondrinker; Light-to-moderate drinker	1 0.5 (0.3–0.7)

IADL: Instrumental Activities of Daily Living; MCI: Mild cognitive impairment; DM-2: Type 2 diabetes mellitus; BMI: Body mass index; TIA: Transient ischemic attack.

**Table 3 ijms-21-02316-t003:** Evidence-based studies on the relationship between alcohol and PD.

Reference, Year	Study Population	Study Design	No. of Subjects (Cases/ Controls)	Diagnosis Criteria and Case Ascertainment	Adjusted Confounders	Alcohol Exposure	OR (95%CI)
Liu et al., 2013 [80]	NIH-AARP Diet and Health Study	Cohort	1113/306,895	PD clinic, medical records	Age, gender, race, education, marital status, smoking, caffeine intake, general health status, physical activity	Beer (drinks/day) 0 <1 1–1.99 ≥2 Wine <1 1–1.99 ≥2 liquor <1 1–1.99 ≥2	1 0.79 (0.68–0.92) 0.73 (0.50–1.07) 0.86 (0.60–1.21) 1.07 (0.92–1.25) 0.74 (0.53–1.02) 1.31 (0.89–1.94) 1.06 (0.91–1.23) 1.22 (0.94–1.58) 1.35 (1.02–1.80)
Palacios et al., 2012 [78]	The Cancer Prevention Study II Nutrition Cohort	Cohort	605/132,403	PD clinic, treating physicians and medical record review	Age, smoking, coffee intake, caloric intake, dairy intake, use of ibuprofen, physical activity and baseline body mass index pesticide exposure, education	Men 0 <9.9 g/day 10–19.9 20–29.9 >30 Women <9.9 g/day 10–19.9 20–29.9 >30	1 1.36 (1.06–1.74) 1.48 (1.09–2.01) 1.15 (0.69–1.90) 1.29 (0.90–1.86) 0.95 (0.68–1.31) 0.95 (0.57–1.60) 1.67 (1.06–2.64) 0.77 (0.41–1.45)
Fukushima et al., 2010 [81]	11 collaborating hospitals in Japan	Case- control	214/327	The UK PD Society Brain Bank clinical diagnostic criteria	Sex, age, region of residence, smoking, education, BMI, alcohol flushing status, presence of medication history for hypertension, hypercholesterolemia, diabetes, caffeine intake, cholesterol, vitamin E, vitamin B6, iron, and dietary glycemic index	0 <6 days per week ≥6 days per week Amount per day (ethanol, g) 0.1–65.9 ≥66.0 Amount per week (ethanol, g) 0.1–219.3 ≥219.4	1 1.29 (0.78–2.13) 0.96 (0.50–1.81) 1.07 (0.64–1.80) 1.46 (0.79–2.71) 0.98 (0.58–1.65) 1.79 (0.95–3.39)
Nicoletti et al., 2010 [82]	Five Movement Disorder centers in Central-Southern Italy	Case- control	492/459	The diagnostic criteria proposed by Gelb et al. in 1999 [83]	Age, sex, family history, place of residence coffee consumption (ever/never) and smoking (ever/never)	Wine (Glasses/day) None (reference) 1–2 3+ Years of wine drinking 1–45 years 46+ years	1 0.68 (0.47–0.97) 0.45 (0.28–0.74) 0.83 (0.55–1.23) 0.45 (0.29–0.68)
Brighina et al., 2009 [84]	Mayo Clinic in Rochester	Case- control	893/893	Incidence and distribution of parkinsonism in Olmsted County [85]	Age and sex, education, smoking, and coffee use	Overall (ever vs. never) Beer Wine Liquor	0.88 (0.68–1.12) 0.96 (0.75–1.23) 1.19 (0.95–1.50) 0.83 (0.67–1.02)
Tan et al., 2008 [86]	Singapore Chinese Health Study; prospective cohort	Cohort	157/63,257	Advisory Council of the US National Institute of Neurological Disorders and Stroke [83]	Age, year of interview, gender, dialect group, education, smoking, tea, coffee, total caffeine intake	Non- or less-than- weekly drinkers At least weekly drinkers	1 0.60 (0.31–1.16)
Dick et al., 2007 [87]	Five European countries	Case- control	767/1989	United Kingdom Parkinson’s Disease Society Brain Bank clinical diagnostic criteria [88]	Age, sex, country, ever used tobacco, ever been knocked unconscious and family history	Ever consumed beer, wine or spirits regularly	0.92 (0.74–1.15)
Wirdefeld et al., 2005 [89]	Cohort (the Swedish Twin Registry)	Co- twin control compa-rison case- control	476/2380	ICD, IDR and CDR	Age, gender, smoking, coffee intake, education, area of living	No alcohol 0–5gm/day 6–15gm/day 16–30gm/day >30gm/day	1 0.72 (0.52–0.99) 1.05 (0.74–1.50) 0.94 (0.52–1.71) 0.66 (0.34–1.29)
Hernán et al., 2004 [90]	The General Practice Research Database (GPRD)	Case- control	1019/10,123	PD clinic, medical records	Age, gender, start date	>500mL/week >0–5 units/week >5–15 >15–30 >30–50 > 50	1.09 (0.67–1.78) 1.10 (0.91–1.33) 1.10 (0.89–1.36) 1.27 (0.96–1.68) 0.57 (0.28–1.18) 1.46 (0.69–3.01)
Hernán et al., 2003 [91]	Nurses’ Health Study, Health ProfessionalsFollow-up Study	Cohort	415/13689	Medical records, NDI	Age, smoking, caffeine intake	0 >0 to <5 gm/day 5 to <15 gm/day 15 to <30 gm/day ≥30 gm/day	1 1.0 (0.8–1.3) 1.0 (0.8–1.4) 1.1 (0.8–1.6) 0.7 (0.5–1.2)
Checkoway et al. 2002 [92]	Western Washington State	Case- control	210/347	PD clinic, medical records, database	Age, ethnicity, education, and gender	Drinks/week 0 1–2 3–9 ≥10	1 1.1 (0.7–1.8) 1.1 (0.6–1.7) 0.8 (0.4–1.4)
Paganini- Hill et al., 2001 [77]	Leisure World Laguna Hills, Southern California	Case- control	395/2320	PD clinic	Age, gender, birthdate, vital status	2+ alcoholic drinks/day	0.77 (0.58–1.03)

AARP: American Association of Retired Persons; ICD: International Classification of Diseases; IDR: Inpatient Discharge Register; CDR: Cause of Death Register; NDI: National Death Index.

**Table 4 ijms-21-02316-t004:** Evidence-based studies on the relationship between alcohol intake and ALS.

Reference, Year	Study Population	Study Design	No. of Subjects Cases/Controls)	Diagnosis Riteria and Case Ascertainment	Adjusted Confounders	Alcohol Exposure	Or (95% ci)
D’Ovidio et al., 2019 [40]	Population-based, Euro-MOTOR study	Case– control	1557/2922	Revised El Escorial Criteria [129], ALS clinic and registry	Sex, age, cohort, education, leisure time physical activity, smoking, heart problems, hypertension, stroke, cholesterol and diabetes	Ever exposed to alcohol Ever exposed to red wine	0.93 (0.75–1.15) 0.99 (0.84–1.16)
Ji et al., 2016 [130]	National cohort in Sweden	Cohort	7965/420,489	ICD, registry	Age at diagnosis, sex, education, birth country and period at diagnosis	Overall Male Female	0.54 (0.45–0.63) 0.52 (0.43–0.63) 0.60 (0.39–0.88)
Huisman et al., 2015 [128]	Population-based, Netherlands	Case- control	674/2093	Revised El Escorial criteria	Age, sex, educational level, BMI, smoking, lifetime physical activity; total energy intake	Higher intake of alcohol	0.91 (0.84–0.99)
Sonja et al., 2012 [127]	Population-based in the Netherlands	Case- control	494/1599	Revised El Escorial Criteria, questionnaire	Age, gender, smoking status, educational level, and alcohol consumption	Never drinker Former drinker Current drinker	1 0.67 (0.40–1.13) 0.52 (0.40–0.75)
Okamoto et al., 2009 [131]	Six medical centers in the Tokai area	Case- control	183/366	El Escorial World Federation of neurology criteria [129]	Age, sex, smoking, bone fracture, vigorous physical activity, stress, Intake of green–yellow vegetables	Nondrinker Current drinker	1 1.1 (0.7–1.5)
Nelson et al., 2000 [132]	Population-based, western Washington State	Case- control	161/321	Medical Examiner	Age, gender, race, smoking status, education	Nondrinker Drinker, ≤2 drinks/day >2 drinks/day	1 0.8 (0.5–1.2) 0.7 (0.3–1.4)
Kamel et al., 1999 [133]	Population-based study in New England	Case- control	109/256	Medical examiner or laboratory supported	Age, sex, region and education, smoking status	Ever used alcohol Drinks per month 5 years ago 0 1–30 >30	1.1 (0.4–3.2) 1 0.9 (0.5–1.8) 1.4 (0.7–3.0)

BMI: Body mass index; ICD: International Classification of Diseases.

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
