# Peer review of "Role of Alcohol Drinking in Alzheimer’s Disease, Parkinson’s Disease, and Amyotrophic Lateral Sclerosis"

_ijms, 2020, doi:10.3390/ijms21072316_

Round 1
Reviewer 1 Report
This manuscript is an overview of the relationship between alcohol consumption and the development of age-related neurodegenerative diseases. Particularly, the authors reported recent epidemiological and experimental evidence regarding the association between alcohol consumption and Alzheimer’s disease (AD), Parkinson's disease (PD) and Amyotrophic Lateral Sclerosis (ALS). They summarized the different existing link between alcohol and the above-mentioned diseases, denoting also the possible dose-dependent effects of alcohol in their development. Interestingly, it appeared the possible dual role of alcohol intake in Alzheimer's disease. Indeed, both epidemiological studies and experimental approaches reported a reduction in the prevalence of Alzheimer's disease in individuals that ingest low amounts of alcohol; differently, excessive amounts of ethanol can be correlated with an increased possibility to develop AD. Furthermore, the authors reported evidence that suggest the alcohol protective effect or the absence of its association with Parkinson’s disease. However, it seems that some experimental animal studies indicate that chronic heavy alcohol consumption may have dopamine neurotoxic effects. Finally, the authors reported that findings in the association between alcohol consumption and amyotrophic lateral sclerosis (ALS) seem inconsistent. However, preliminary studies suggested that drinking alcohol seems to have no influence on the risk of developing amyotrophic lateral sclerosis (ALS). To conclude the authors underlined the importance of more research to clarify the potential involvement of alcohol intake in causing neurodegenerative diseases.
The topic of the manuscript is interesting, and the goal is to provide an update of the available data on the connection between the alcohol intake and the risk to develop AD, PD, and ALS. The paper is well written, and the overall presentation is sufficiently clear. The literature review is recent, complete and exhaustive. However, for the manuscript to be accepted minor revisions are required. The authors should carefully revise their text to correct incongruencies and careless mistakes throughout the text (as under the “Other issues” section below). In addition, the figures in the manuscript should be strongly revisioned.
Other issues:
Line 89: Lipopolysaccharide
Lines99: one recent study showed the hyperphosphorylation..
Lines from 99 to 101 as well as from 200 to 201: the authors assert that “Tau is hyperphosphorylated in 3xTG-AD mice after 1-month post alcohol drinking compared to control”. The authors should report the mice condition treatment, in other words, they should explain if 3xTg-AD mise were treated for 1 month with high or low levels of alcohol, considering that they support the idea that low or high alcohol intake can differently affect the development of AD.
Line 102: “Alcohol has been suggested to be either protective of, or not associated with Parkinson’s disease.” I think that a reference to support this affirmation is needed.
Line 117: The figure does not really represent the dual role of alcohol in AD, PD and, ALS. This can be said only for AD but not for PD and ALS where, the presence of only one arrow, underline that the potential dual role of alcohol is missing. In this respect the authors should replace “Dual roles of ..“ with “Relationship between Alcohol and AD, PD and ALS” ;
Line 119: the authors report a reference number ([35]) into the figure explanation. They should decide to insert all references also for the rest of the sentences in the figure explanation or remove that one.
Line 120: the authors should correct “red wine” with “red wine extract" as reported in line 109 since their meaning are completely different then, it is not correct.
Line 263-264: the authors should introduce a reference for this sentence.
Line 271: “CYP2E1 knockout mice treated with neurotoxin”
Line 273: “brain” without s
Line 119: the authors report “in PD, most studies showed that alcohol decreased levels of DA and increased the levels of alphaSyn.” Even in this case, to validate this affirmation, the authors should provide more information about alcohol exposure, if under high or low levels.
Lines 360: excitatory system in ALS, and chronic alcohol…
Line 364: In this consideration, it is likely that alcohol..
Line 365: ,we can not concluded..
Figure 1: remove the grammar correction sign below “vagus” and “iNOS”
Figure 2: the dual role of alcohol in AD, PD and, ALS is incomprehensible. Only for AD appears the dual role of alcohol depending on its amount, denoted with two different arrows. For PD is unclear if the effect on DA and Alphasyn is due to high or low levels of alcohol intake. For ALS is clear that the “dual” role of alcohol must be better investigate. Thus, the authors should reorganize the figure.
Author Response
Point 1: This manuscript is an overview of the relationship between alcohol consumption and the development of age-related neurodegenerative diseases. Particularly, the authors reported recent epidemiological and experimental evidence regarding the association between alcohol consumption and Alzheimer’s disease (AD), Parkinson's disease (PD) and Amyotrophic Lateral Sclerosis (ALS). They summarized the different existing link between alcohol and the above-mentioned diseases, denoting also the possible dose-dependent effects of alcohol in their development. Interestingly, it appeared the possible dual role of alcohol intake in Alzheimer's disease. Indeed, both epidemiological studies and experimental approaches reported a reduction in the prevalence of Alzheimer's disease in individuals that ingest low amounts of alcohol; differently, excessive amounts of ethanol can be correlated with an increased possibility to develop AD. Furthermore, the authors reported evidence that suggest the alcohol protective effect or the absence of its association with Parkinson’s disease. However, it seems that some experimental animal studies indicate that chronic heavy alcohol consumption may have dopamine neurotoxic effects. Finally, the authors reported that findings in the association between alcohol consumption and amyotrophic lateral sclerosis (ALS) seem inconsistent. However, preliminary studies suggested that drinking alcohol seems to have no influence on the risk of developing amyotrophic lateral sclerosis (ALS). To conclude the authors underlined the importance of more research to clarify the potential involvement of alcohol intake in causing neurodegenerative diseases.
The topic of the manuscript is interesting, and the goal is to provide an update of the available data on the connection between the alcohol intake and the risk to develop AD, PD, and ALS. The paper is well written, and the overall presentation is sufficiently clear. The literature review is recent, complete and exhaustive. However, for the manuscript to be accepted minor revisions are required. The authors should carefully revise their text to correct incongruencies and careless mistakes throughout the text (as under the “Other issues” section below). In addition, the figures in the manuscript should be strongly revisioned.
Response 1: Thanks for reviewer 1’s overall enthusiasm and comment with minor revision.
Below please find our responses under “Other issues” section on a point by point basis.
Point 2: Line 89: Lipopolysaccharide
Response 2: we have changed “lipopolysaccharide” into “Lipopolysaccharide” and “peptidoglycan” into “Peptidoglycan”.
Point 3: Lines99: one recent study showed the hyperphosphorylation.
Lines from 99 to 101 as well as from 200 to 201: the authors assert that “Tau is hyperphosphorylated in 3xTG-AD mice after 1-month post alcohol drinking compared to control”. The authors should report the mice condition treatment, in other words, they should explain if 3xTg-AD mise were treated for 1 month with high or low levels of alcohol, considering that they support the idea that low or high alcohol intake can differently affect the development of AD.
Response 3: we have changed “One recent study showed that hyperphosphorylation of Tau-Ser199/202 in neuronal cell bodies and dorsal hippocampus (CA1) in 3xTg-AD mice after 1-month post alcohol drinking compared to controls” into “One recent study showed that alcohol drinking in the triple-transgenic model of AD (3xTg-AD) mice induced deficit in cognitive and emotional functions as compared to wild-type controls, as well as induced pathological changes such as the hyperphosphorylation of Tau-Ser199/202 in neuronal cell bodies and the dorsal hippocampus (CA1) in 3xTg-AD mice after 1-month post alcohol drinking.”
Point 4: Line 102: “Alcohol has been suggested to be either protective of, or not associated with Parkinson’s disease.” I think that a reference to support this affirmation is needed.
Response 4: we have inserted a new reference “[36]”.
Point 5: Line 117: The figure does not really represent the dual role of alcohol in AD, PD and, ALS. This can be said only for AD but not for PD and ALS where, the presence of only one arrow, underline that the potential dual role of alcohol is missing. In this respect the authors should replace “Dual roles of ..“ with “Relationship between Alcohol and AD, PD and ALS” ;
Response 5: we have modified Figure 2 accordingly.
Point 6: Line 119: the authors report a reference number ([35]) into the figure explanation. They should decide to insert all references also for the rest of the sentences in the figure explanation or remove that one.
Response 6: we have inserted all references in the legend of Figure 1.
Point 7: Line 120: the authors should correct “red wine” with “red wine extract" as reported in line 109 since their meaning are completely different then, it is not correct.
Response 7: we have changed “red wine” into “red wine extract".
Point 8: Line 263-264: the authors should introduce a reference for this sentence.
Response: we have inserted a reference “[99]”.
Point 9: Line 271: “CYP2E1 knockout mice treated with neurotoxin”
Response 9: we have changed “CYP2E1 knockout mice with neurotoxin” into “CYP2E1 knockout mice treated with neurotoxin”.
Point 10: Line 273: “brain” without s
Response 10: we have changed “brains” into “brain”.
Point 11: Line 119: the authors report “in PD, most studies showed that alcohol decreased levels of DA and increased the levels of alphaSyn.” Even in this case, to validate this affirmation, the authors should provide more information about alcohol exposure, if under high or low levels.
Response 11: in the revised manuscript, we added “One study presented that the high drinking alkoalcohol (AA) rats showed lower stimulated DA levels and more efficient DA re‐uptake in the nucleus accumbens core than the low drinking non‐alcohol (ANA) rats did. In the same structure, AA rats had also significantly higher levels of α‐syn. No differences were found in the effects of two different doses of ethanol (0.1 and 3.0 g/kg)”.
Point 12: Lines 360: excitatory system in ALS, and chronic alcohol…
Response 12: we have changed “excitatory system in ALS. And chronic alcohol” into “excitatory system in ALS, and chronic alcohol”.
Point 13: Line 364: In this consideration, it is likely that alcohol..
Response 13: we have modified this sentence and changed into “In this consideration, it is likely that alcohol”.
Point 14: Line 365: ,we can not concluded..
Response 14: we have changed “we can’t concluded” into “we cannot concluded”
Point 15: Figure 1: remove the grammar correction sign below “vagus” and “iNOS”
Response 15: the grammar correction sign below “vagus” and “iNOS” have been removed.
Point 16: Figure 2: the dual role of alcohol in AD, PD and, ALS is incomprehensible. Only for AD appears the dual role of alcohol depending on its amount, denoted with two different arrows. For PD is unclear if the effect on DA and Alphasyn is due to high or low levels of alcohol intake. For ALS is clear that the “dual” role of alcohol must be better investigate. Thus, the authors should reorganize the figure.
Response 16: thanks for the constructive criticism and we have modified Figure 2.
Reviewer 2 Report
The paper has adequately reviewed about the impact of alcohol on neuro-degenerative diseases.
- However, the syntax in the sentences pertaining to "Abstract" and "Introduction" can be improved.
- Few abbreviations need to be explained at their first use:
- line 34 : CYP2E1
- line 62: APOE-e4
- line 91-93
- line 101: 3xTg-AD
Apart from reviewing extensively, the syntax errors for Abstract and Introduction: Few sentences need revising:
- line 131: "Some factors are nonmodifiable risk factors, including....." can be , for example, "Some factors are non-modifiable like......."
- line 211: "The prevalence ...... predominance." The last words "male predominance" do not correctly fit in with the initial part of the sentence.
Author Response
Point 1: The paper has adequately reviewed about the impact of alcohol on neuro-degenerative diseases.
However, the syntax in the sentences pertaining to "Abstract" and "Introduction" can be improved.
Response 1: Thanks for reviewer 2’s consideration and constructive comments. We have carefully re-edited the English of this revised manuscript.
Point 2: Few abbreviations need to be explained at their first use:
line 34 : CYP2E1
line 62: APOE-e4
line 91-93
line 101: 3xTg-AD
Response 2: we have added the full-name of each abbreviated terminology accordingly.
Point 3: Apart from reviewing extensively, the syntax errors for Abstract and Introduction: Few sentences need revising:
Response: A few sentences have been revised.
Point 4: line 131: "Some factors are nonmodifiable risk factors, including....." can be , for example, "Some factors are non-modifiable like......."
Response 4: we have changed “Some factors are nonmodifiable risk factors, such as…..” into
“Some factors are nonmodifiable like….”.
Point 5: line 211: "The prevalence ...... predominance." The last words "male predominance" do not correctly fit in with the initial part of the sentence.
Response 5: we re-edited the sentence as “The prevalence and incidence of PD increase with age and males show predominance”.